# A loss-of-adhesion CRISPR-Cas9 screening platform to identify cell adhesion-regulatory proteins and signaling pathways

Martin F. M. de Rooij[1,2,3,6], Yvonne J. Thus[1,2,3,6], Nathalie Swier[1,2,3], Roderick L. Beijersbergen [4,5], Steven T. Pals[1,2,7] & Marcel Spaargaren [1,2,3,7 ✉]

The clinical introduction of the Bruton's tyrosine kinase (BTK) inhibitor ibrutinib, which targets B-cell antigen-receptor (BCR)-controlled integrin-mediated retention of malignant B cells in their growth-supportive lymphoid organ microenvironment, provided a major breakthrough in lymphoma and leukemia treatment. Unfortunately, a significant subset of patients is intrinsically resistant or acquires resistance against ibrutinib. Here, to discover novel therapeutic targets, we present an unbiased loss-of-adhesion CRISPR-Cas9 knockout screening method to identify proteins involved in BCR-controlled integrin-mediated adhesion. Illustrating the validity of our approach, several kinases with an established role in BCR-controlled adhesion, including BTK and PI3K, both targets for clinically applied inhibitors, are among the top hits of our screen. We anticipate that pharmacological inhibitors of the identified targets, *e.g.* PAK2 and PTK2B/PYK2, may have great clinical potential as therapy for lymphoma and leukemia patients. Furthermore, this screening platform is highly flexible and can be easily adapted to identify cell adhesion-regulatory proteins and signaling pathways for other stimuli, adhesion molecules, and cell types.

[1] Department of Pathology, Amsterdam UMC location University of Amsterdam, Meibergdreef 9, Amsterdam, The Netherlands. [2] Lymphoma and Myeloma Center Amsterdam (LYMMCARE), Amsterdam, The Netherlands. [3] Cancer Center Amsterdam (CCA), Cancer Biology and Immunology - Target & Therapy Discovery, Amsterdam, The Netherlands. [4] Division of Molecular Carcinogenesis and Oncode Institute, The Netherlands Cancer Institute, Plesmanlaan 121, Amsterdam, The Netherlands. [5] NKI Robotics and Screening Center and Genomics Core Facility, The Netherlands Cancer Institute, Plesmanlaan 121, Amsterdam, The Netherlands. [6] These authors contributed equally: Martin F.M. de Rooij, Yvonne J. Thus. [7] These authors jointly supervised this work: Steven T. Pals, Marcel Spaargaren. ✉email: marcel.spaargaren@amsterdamumc.nl

Over the past decade, the introduction of B cell antigen receptor (BCR) signalosome inhibitors—viz. Bruton's tyrosine kinase (BTK) and PI3K inhibitors—provided a major breakthrough in the treatment of B cell malignancies, with objective response rates of 70–90%[1–8]. In many B cell malignancies, such as chronic lymphocytic leukemia (CLL), mantle cell lymphoma (MCL), and Waldenström macroglobulinemia (WM), these drugs are not directly cytotoxic, but rather result in mobilization of the malignant B cells from their protective lymphoid organ microenvironment into the circulation, resulting in cell death due to lack of growth and survival signals[2–7,9–13].

The retention of CLL, MCL, and WM cells in the lymphoid environment is mediated by integrin adhesion molecules, which are controlled by BCR signaling. BCR signaling in normal and malignant B cells is activated by cognate antigens presented in lymphoid tissues, and evokes proliferation, survival, and differentiation signals, as well as retention in lymphoid tissues via focal adhesion complex formation. Previously, we and others demonstrated that targeting of BCR-controlled integrin-mediated adhesion underlies the clinical efficacy of BCR-signalosome inhibitors in CLL, MCL, and WM[9–13].

The currently most common clinically applied BCR-signalosome inhibitors are the BTK inhibitors ibrutinib, acalabrutinib, and zanubrutinib. Upon prolonged treatment, however, a significant subset of patients develop resistance against these inhibitors[14–16], and after discontinuation of ibrutinib CLL patients have a median overall survival of only 3 months[17]. Most often the drug resistance is caused by acquired mutations in BTK or its substrate PLCγ2[16,18–25], demonstrating that BCR-signaling (and probably the BCR–integrin axis) is not easily bypassed by other pathways in these B cell malignancies. Although the PI3Kδ inhibitors idelalisib (δ), duvelisib (γδ), and copanlisib (αδ) also display good clinical activity in B cell malignancies, especially the first two drugs could cause severe adverse effects[26]. Thus, there is an unmet need for novel druggable targets.

Unbiased CRISPR-Cas9 knockout screens are a great added value in cancer research to find vulnerabilities for specific tumor types, overcome drug resistance, and find synthetic lethal interactions. Thus far, most of these functional screens have assessed biological effects on cell growth and viability. Here, we present the development of a loss-of-adhesion CRISPR-Cas9 knockout screening method to identify targets involved in BCR-controlled integrin-mediated adhesion of malignant B cells. Demonstrating the validity and quality of the screen, several known targets for BCR-controlled integrin-mediated adhesion, such as BTK, PI3K, and SYK were among the top 10 hits of our screen. Our applied CRISPR screen provides insights in BCR-controlled regulation of integrin-mediated adhesion of (malignant) B cells and, moreover, identifies several targets for lymphoma therapy. Furthermore, the developed screening platform can be easily adapted for functional genomic studies aimed at the identification of cell adhesion-regulatory proteins for other stimuli, adhesion molecules, and cell types.

## Results

**Generation of stable Cas9-expressing cells.** For the loss-of-adhesion CRISPR screen, we employed the B cell line Namalwa, since of critical importance given the nature of our screen, this cell line displays excellent BCR-stimulated integrin-mediated adhesion to fibronectin and, unlike many other B cell lines, does not depend on BCR signaling for cell growth and survival[10,27]. To achieve an optimal transduction efficiency of the CRISPR-guide library, we used a dual vector system, including lentiCas9 and a lentiGuide-derived library[28]. Upon transduction with LentiCas9, selected Namalwa clones were analyzed for optimal BCR-stimulated adhesion capacity (Supplementary Fig. 1a), and for

Cas9 expression and activity, as determined by means of a lentiviral Cas9-reporter[29] (Supplementary Fig. 1b–d). Cas9 was active in more than 99% of the cells of clone #10, which is crucial for pooled screening.

**BCR-controlled loss-of-adhesion CRISPR screen with a kinome-centered CRISPR-Cas9 library.** To identify kinases involved in the antigen/BCR-integrin axis, we performed a loss-of-adhesion CRISPR screen with the lentiviral Brunello kinome-(like)-centered CRISPR-guide library. The library was transduced in the selected Cas9-expressing Namalwa clone (clone #10) with a multiplicity of infection of 0.3 and a representation of at least 1000 times the library size (Fig. 1a). After puromycin selection for 2 days, the transduced cells were cultured until the viability was restored to >90% (6 more days). Immediately before the adhesion experiment, a pre-adhesion sample was taken. For the adhesion screen we used fibronectin as integrin ligand since adhesion to fibronectin is mediated by both α4- and α5-integrins whereas adhesion to vascular cell adhesion molecule 1 (VCAM-1) is limited to α4-integrins, and since VCAM-1, unlike fibronectin, has a very narrow concentration window for optimal adhesion. Cells in one experimental arm were stimulated with αIgM (which mimics antigen-induced BCR ligation) and allowed to adhere to fibronectin-coated wells. In a second arm, cells were stimulated with phorbol-12-myristate-13-acetate (PMA), a diacylglycerol analog that activates PKC distally from the BCR signalosome (see Fig. 2a), and were also allowed to adhere to fibronectin. After removal of non- or weakly adhered cells by extensive washing, i.e., until unstimulated cells in control wells were completely removed, the adhered cells were harvested by EDTA-mediated detachment (integrin-mediated adhesion is calcium and magnesium dependent).

Quantification of the guides in the pre- and post-adhesion samples was performed by employing a barcoded PCR on genomic DNA, followed by next generation sequencing[30,31]. From the reads, the barcodes and guide sequences were identified and their abundance was determined. The DESeq2 pipeline[32] was applied to obtain statistics at the guide level from the count table. This was followed by αRRA (from the MAGeCK pipeline[33]) to obtain statistics at the gene level from the guide statistics (Fig. 1a). The screen was performed with 3 independent replicates, in which the pre-adhesion, αIgM-induced adhesion, and PMA-induced adhesion samples were paired. The R-squared between the read counts of the independent replicates of every condition was 85.5–92.2% (Supplementary Fig. 2a), demonstrating high reproducibility and excellent maintenance of the library complexity.

**Validation of the screen by negative and positive control genes, including established mediators of BCR-controlled integrin activation.** To address the performance of our screen, we first determined the behavior of negative control genes. As negative controls, by analogy to the non-essential genes as defined by Hart et al. for functional genomic lethality screens[34], we employed genes which are represented in the guide library but are not expressed in Namalwa (Supplementary Fig. 2b). In the MA plots, the distribution of these negative control guides falls within the cloud with fold changes typically between -0.5 and 0.5 on a log2 scale (Fig. 1b). Genes involved in cell viability were identified by comparing read counts of the pre-adhesion samples with the distribution of the library (Supplementary Fig. 2c). The cell line-specific controls behave similar to the (limited available) Hart-defined essential and non-essential controls[34] (Supplementary Fig. 2c, d). As expected, the viability genes strongly affected both αIgM and PMA-stimulated adhesion (Fig. 1b), as nonviable cells will not adhere well.

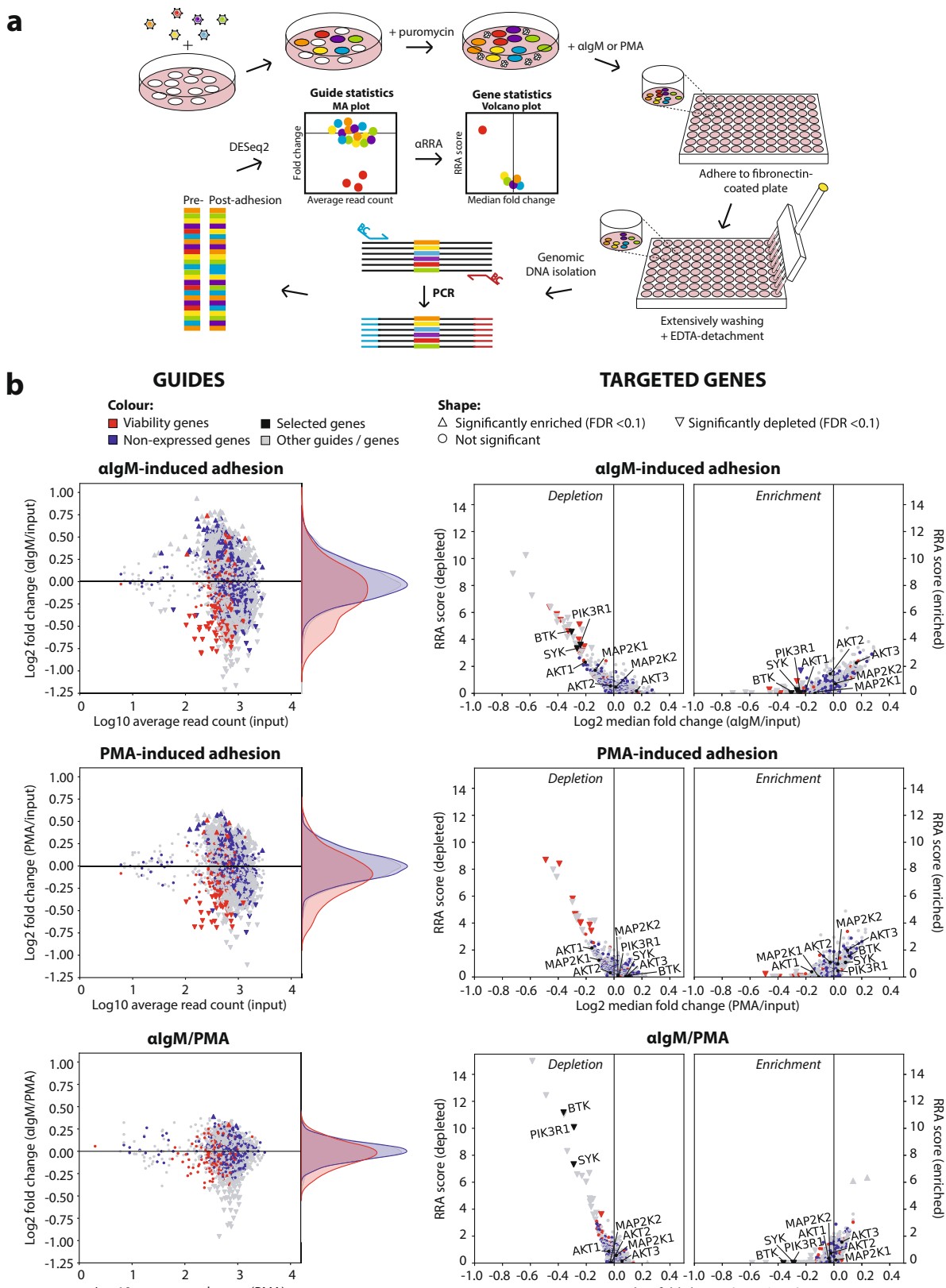

By comparing the αIgM- with the PMA-stimulated arm, the "noise" caused by the guides that reduce viability was strongly diminished (Fig. 1b bottom panel). Importantly, in this comparison BTK, PI3K, and SYK, which represent established crucial kinases of the BCR signalosome[9–13,27] (Fig. 2a), ranked within the top 10 of depleted genes (see Fig.4a); they were clearly

separated from the cloud in the volcano plot (Fig. 1b), and had a FDR <0.1. In sharp contrast, guides targeting MEK and AKT, kinases which have previously been demonstrated not to be involved in the antigen/BCR-integrin axis[10,27] (Fig. 2a), were not depleted (Fig. 1b). The ability (or disability) to inhibit αIgM-induced adhesion by pharmacological inhibitors against these

**Fig. 1 Loss-of-adhesion CRISPR screen for BCR-controlled integrin-mediated adhesion of lymphoma cells. a** Experimental set-up of the loss-of-adhesion CRISPR screen. Namalwa cells were transduced with the Brunello kinome-centered library at MOI of 0.3 and representation of >1000 times the library size. Upon puromycin selection, the cells were grown till viability was recovered (>90%). Cells were stimulated with αIgM or PMA, and allowed to adhere to fibronectin-coated surfaces. Upon extensively washing, the strongly adhered cells were detached with EDTA. From these cells, genomic DNA was isolated, and the presence of the guides was determined by a barcoded PCR and Illumina next generation sequencing (NGS). With DESeq2 and αRRA (from MAGeCK), statistics was obtained on guide and gene level respectively. See online method section for details. The model is created by MFM de Rooij. **b** Namalwa screen results on guide and gene level. Left: MA plots, with the average normalized read count of reference samples (*x*-axis) versus the fold change (*y*-axis); right: volcano plots, with the median fold change of the guides per gene (*x*-axis) versus the $\alpha RRA_{depletion}/\alpha RRA_{enrichment}$ scores (*y*-axis) of either αIgM-induced adhesion relative to pre-adhesion (top), PMA-induced adhesion relative to pre-adhesion (center), and αIgM-induced adhesion relative to PMA-induced adhesion (bottom). The screen was performed with 3 independent replicates. Source data are provided as a Source data file.

targets nicely reflects the depletion (or not) of the guides targeting the corresponding genes (Fig. 2b). Notably, we identified *PIK3R1*, which encodes the p85 regulatory subunit for the catalytic PI3K subunits p110α and p110δ, as a positive regulator of adhesion. In contrast, p110α or p110δ were not significantly depleted in the αIgM/PMA comparison, indicating functional redundancy in Namalwa. This is in agreement with our results with pharmacological inhibitors: the specific PI3Kδ inhibitor idelalisib did not inhibit BCR-controlled adhesion in Namalwa, in contrast to the inhibition observed in the MCL cell line JeKo-1, but the pan-PI3K inhibitor wortmannin did inhibit adhesion (Supplementary Fig. 3).

**Identification of other proteins involved in BCR signaling**. Upon BCR ligation, LYN mediates activation signals, such as phosphorylation of the CD79A/B ITAM motives, SYK, and BTK (Fig. 2a). However, it also instigates negative feedback signals, by phosphorylating immunoreceptor tyrosine-based inhibitory motives of inhibitory receptors, resulting in recruitment of phosphatases such as SHIP and SHP1/2[35]. In our screen, guides targeting LYN were not significantly depleted (Fig. 3), suggesting that the activating signal by LYN is redundant. Consistent with this notion, whereas loss-of-function mutations of BTK result in immunodeficiency, loss-of-function mutation of LYN rather results in autoimmunity[36]. The tyrosine kinase CSK, an important negative regulator of SRC family kinases such as LYN (Fig. 2a)[37], dropped out significantly stronger in the αIgM arm than in the PMA arm (Fig. 3). This suggests that negative feedback regulation of LYN by CSK could be important for BCR-controlled adhesion. Interestingly, CSK is also targeted by ibrutinib with an IC$_{50}$ in the low nanomolar range[38].

Guides which target *PRKCB*, *ACTR2*, and *GUK1* were significantly depleted upon both αIgM- as well as PMA-stimulated adhesion (Fig. 3). Indeed, the proteins encoded by these genes function more distally in the antigen/BCR-integrin axis (Fig. 2a). The β-isoform of PKC (*PRKCB*) was involved in both αIgM- and PMA-induced adhesion, while the ε-isoform of PKC (*PRKCE*) was more important for αIgM than for PMA-induced adhesion (Fig. 3). ARP2/3 (*ACTR2*) is involved in actin polymerization, which is important for the focal adhesion complex, and GUK1 is involved in GTP biosynthesis, which is important for integrin-regulatory GTPases such as RAP1, RHO, and RAC (Fig. 2a). However, these genes have less clinical potential because of their ubiquitous expression. Furthermore, in long-term lethality screens knockout of *GUK1* and *ACTR2* was lethal in all our tested cell lines and in public CRISPR screen data from depmap.org.

**Identification of regulators of the BCR–integrin axis**. To identify proteins involved in the (proximal) BCR–integrin axis, we selected the genes with a FDR < 0.1 in the αIgM/PMA comparison. We called 20 significant hits, of which 4 genes (*PAK2*, *PKM*,

*CRKL*, and *PTK2B*) with similar or even better fold changes as the positive control genes (*BTK*, *PIK3R1*, and *SYK*) (Fig. 4a and Supplementary Fig. 4). Importantly, the dropout of the guides against these genes is not due to compromised cell viability (Supplementary Fig. 5). To visualize their specific effect on αIgM-controlled adhesion and not PMA-controlled adhesion, we plotted the median fold changes from αIgM- versus PMA-controlled adhesion: the 20 genes were out of the general cloud and at a significant distance below the diagonal (Fig. 4b). From the above-mentioned 4 top genes we cloned the best 2 guides, based upon drop-out in the screen, to validate their effect on αIgM- and PMA-controlled adhesion in Namalwa. Gene knockout was confirmed by western blot analysis (supplementary Fig. 6a). In general, a good correlation was observed between the individual fold changes of the guides from the CRISPR screen and the relative adhesion levels of the validation experiments (Fig. 4c and Supplementary Fig. 6b). Only the validation of *PKM* (pyruvate kinase M1/2) was divergent from the screen, most likely caused by direct cytotoxicity due to impaired glycolysis, resulting in lower adhesion levels upon PMA-stimulation than expected, and by competitive outgrowth of *PKM*-proficient cells (due to in frame indels or subclonally insufficient CRISPR-Cas9 activity), resulting in higher adhesion levels upon αIgM-stimulation than expected.

Next, we investigated the 4 top hits in JeKo-1 cells, an MCL cell line with excellent BCR-controlled adhesion (Fig. 4d and Supplementary Fig. 6a, b). Since knockout of *PAK2* and *PKM* was toxic in JeKo-1, PMA-induced adhesion was also affected and knockout efficacy was sub-optimal, due to selective outgrowth of proficient cells (Supplementary Fig. 6a, b). Therefore, to further investigate the role of *PAK2*, we employed the pharmacological PAK2 inhibitors AZ13705339 and FRAX597. In Namalwa, AZ13705339 inhibited αIgM-controlled adhesion and not PMA-induced adhesion, similar to the effect of CRISPR-mediated *PAK2* knockout and ibrutinib treatment, and FRAX597 also inhibited αIgM-controlled adhesion, combined with weak toxicity, whereas both PAK-inhibitors did not affect αIgM-controlled adhesion of JeKo-1 cells (Supplementary Fig. 6b). A hit that was well conserved in the BCR-integrin axis of JeKo1 was *CRKL*: knockout of *CRKL* was very efficient, not toxic, and αIgM-controlled adhesion was strongly impaired, while PMA-controlled adhesion was not affected (Fig. 4d and Supplementary Fig. 6a, b). In addition, knockout of *PTK2B* (also known as PYK2 or FAK2) was also not toxic in JeKo-1, and reduced αIgM-controlled adhesion weakly without affecting PMA-stimulated adhesion (Fig. 4d and Supplementary Fig. 6b). Furthermore, *PTK2B* was found to play a prominent role in BCR-controlled adhesion of the WM cell line BCWM.1 (Supplementary Fig. 6c). Finally, we investigated a significant hit from the screen with a small fold change, the lipid kinase *PIKFYVE*, employing a pharmacological PIKFYVE inhibitor YM201636. As expected, pretreatment of Namalwa, but also JeKo-1, with YM201636 showed a small decrease in αIgM-controlled adhesion,

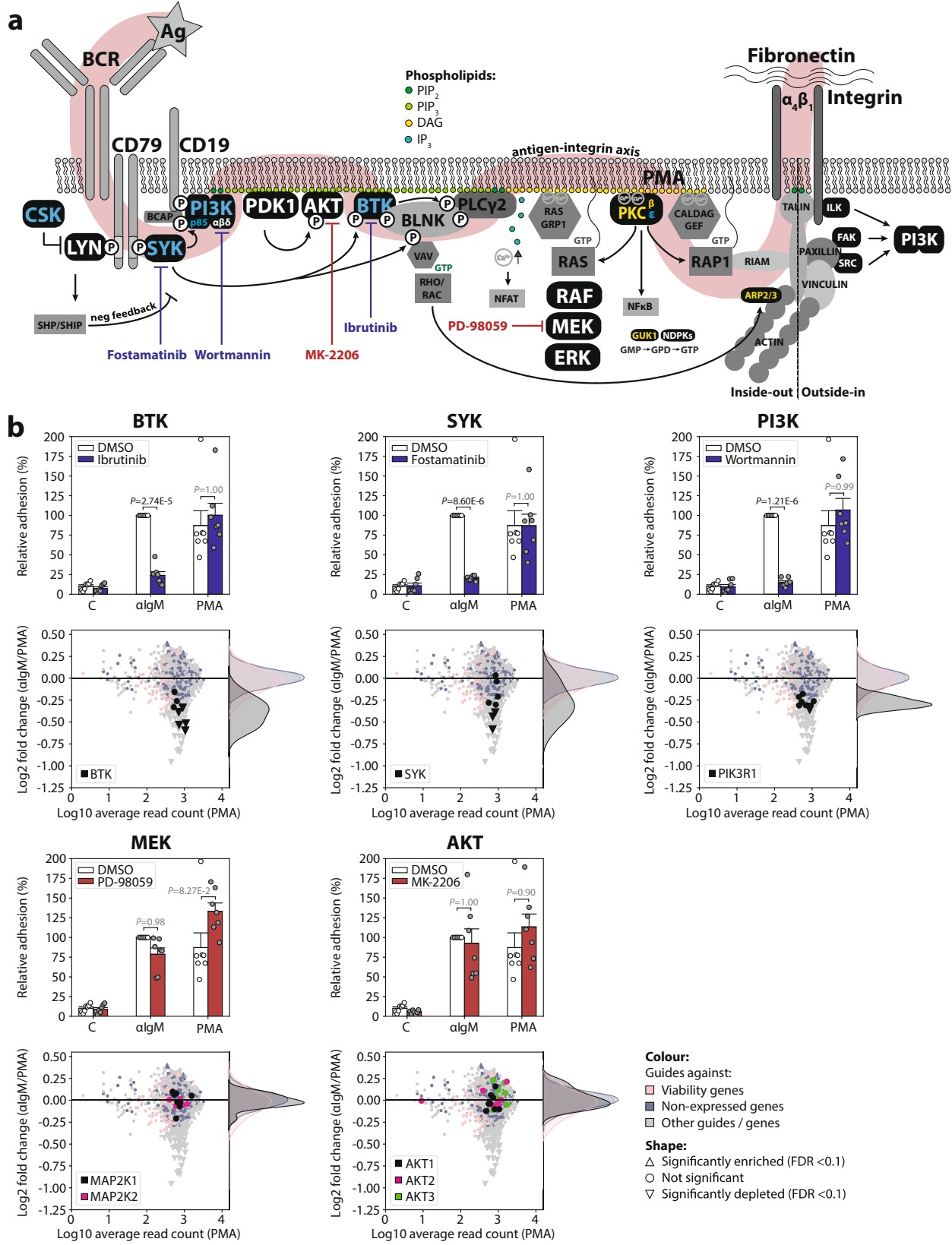

while PMA-controlled adhesion was unaffected (Fig. 4d and Supplementary Fig. 6b). Taken together, these pharmacological and CRISPR-based studies demonstrate that the CRISPR screen adequately identifies proteins with a role in BCR-controlled integrin-mediated adhesion.

## Discussion

In this study, we have developed an unbiased CRISPR screening platform for the identification of positive and negative cell adhesion-regulatory proteins and signaling pathways, and we have successfully conducted a loss-of-adhesion CRISPR screen to identify

**Fig. 2 Validation of the adhesion screen by targeted inhibition of BCR-controlled integrin-mediated adhesion. a** The BCR signaling pathway. Proteins represented by guides in the Brunello library are shown in black. The guides corresponding to the kinase(-related) genes encoding proteins in yellow letters were significantly depleted in both αIgM-induced adhesion and PMA-induced adhesion, the guides corresponding to the proteins in blue letters were significantly depleted in αIgM-induced over PMA-induced adhesion, and the guides corresponding to proteins in white letters were significantly depleted in neither ones. Ag antigen (αIgM), BCR B cell receptor, PIP$_2$ phosphatidylinositol-4,5-diphosphate, PIP$_3$ phosphatidylinositol-3,4,5-triphosphate, DAG diacylglycerol, IP$_3$ inositol-1,4,5-triphosphate, Ca$^{2+}$ cytosolic calcium ions, PMA phorbol-12-myristate-13-acetate, GMP/GDP/GTP guanosine-mono/di/tri-phosphate. The model is created by MFM de Rooij. **b** Comparison of pharmacological inhibitors with the corresponding guides of positive and negative controls for BCR-controlled adhesion. Top plots: Namalwa cells pretreated with 100 nM ibrutinib (BTK inhibitor), 1 µM fostamatinib (SYK inhibitor), 100 nM wortmannin (pan-PI3K inhibitor), 50 µM PD-98059 (MEK1/2 inhibitor), 2.5 µM MK-2206 (AKT1/2/3 inhibitor), or DMSO control, were stimulated with αIgM or PMA, and allowed to adhere to fibronectin-coated surfaces. Bar graphs are presented as normalized mean + SEM (100% = untreated αIgM-stimulated cells). *P* values are adjusted *P* values from two-way ANOVA followed by Tukey HSD post hoc test (two-tailed and paired design), of 7 independent experiments each assayed in triplicate. Bottom plots: MA plots of αIgM-induced adhesion relative to PMA-induced adhesion; the guides of the corresponding genes targeted by the inhibitors of the top plots are highlighted. The screen was performed with three independent replicates. Statistics was performed with DESeq2. Source data are provided as a Source data file.

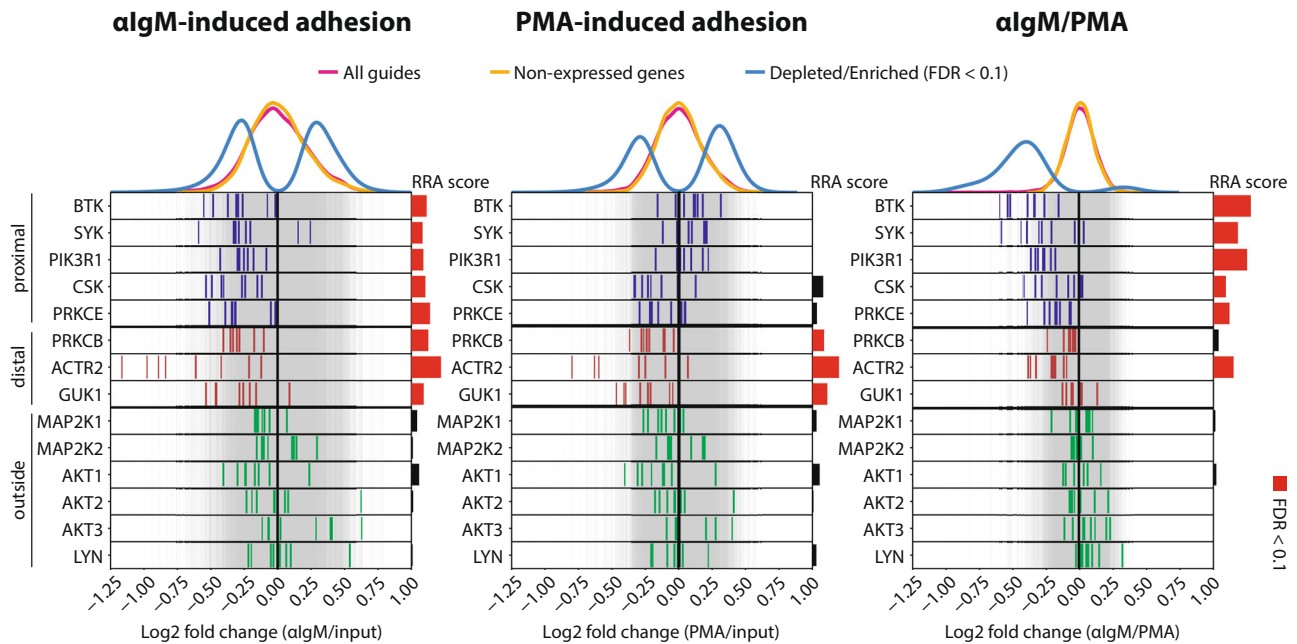

**Fig. 3 Distribution of the guides targeting genes involved in either proximal-, distal-, or not involved in, antigen/BCR-integrin signaling.** Log2 fold changes of the individual guides targeting genes involved in either proximal (blue lines), distal (red lines), or not involved in (green lines) BCR-integrin signaling. The RRA scores are the αRRA$_{depletion}$ scores. The graphs show the results for αIgM-induced adhesion relative to pre-adhesion (left), PMA-induced adhesion relative to pre-adhesion (middle), and αIgM-induced adhesion relative to PMA-induced adhesion (right). Source data are provided as a Source data file.

kinases involved in the antigen/BCR-integrin axis. Our screen was of high quality, since we identified all kinases known to be involved in the antigen/BCR-integrin axis, both proximal kinases, such as BTK, PI3K, and SYK, as well as distal kinase(-related) proteins, such as PKC and ARP2/3 (involved in actin polymerization)[1,27]. Note that ARP2/3 is not a kinase complex, but ARP2 contains an ATP-binding domain for its ATPase activity. We have demonstrated that pharmacological inhibition of these kinases impairs BCR-controlled integrin-mediated adhesion in vitro, in line with the clinically observed lymphocytosis in CLL, MCL, and WM patients, reflecting the mobilization of the malignant cells from their protective lymphoid organ microenvironment into the circulation, resulting in deprivation of critical growth- and survival-factors, followed by lymphoma regression[2–7,9–13].

Apart from known BCR signalosome proteins, we identified several regulators of the BCR-integrin axis. The most significant hit in proximal BCR-signaling (αIgM/PMA) was *PAK2* (p21-Rac1 activated kinase 2). PAK2 acts downstream of RAC1 and is involved in actin remodeling[39], and might be involved in the

BLNK–VAV–RAC1–PAK2–paxilin axis. Since *PAK2* knockout or pharmacological inhibition did not affect BCR-controlled adhesion in every B cell line, this signaling arm appears to be a modifier pathway rather than a canonical pathway. The second hit was *PKM* (pyruvate kinase M1/2), which is involved in glycolysis. Knockout of the glycolytic hits *PKM* and *PGK1* (rank #9), which are responsible for ATP production in the cell, only affected αIgM-controlled adhesion, whereas knockout of *GUK1*, which is involved in GTP production, affected both αIgM as well as PMA-controlled adhesion. This would support a dominant role for small GTPases, which require GTP, rather than protein kinases, which require ATP, downstream of PKC (Fig. 2a). Since both GTP and ATP are also important components in RNA synthesis and metabolism, knockout of *GUK1*, *PKM*, and *PGK1* will be highly lethal on the long-term; hence, the role of these genes in adhesion would be hard to validate by CRISPR-mediated knockout, but they also have low clinical potential. Additional top hits from the Namalwa CRISPR screen were the adapter molecule *CRKL* (CRK-like proto-oncogene) and the

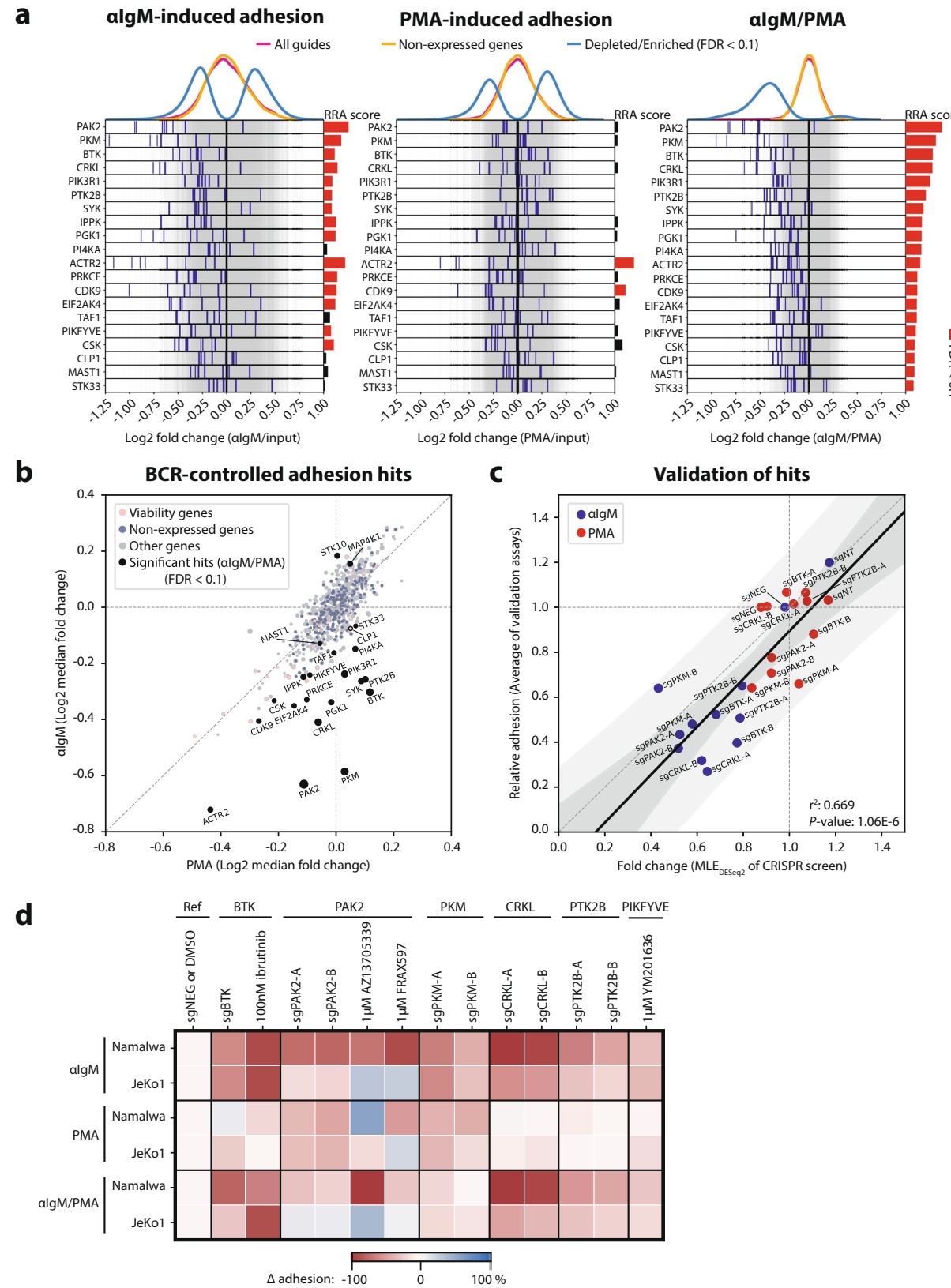

non-receptor protein tyrosine kinase *PTK2B* (aka PYK2 and FAK2). Both proteins are associated with RAP1 signaling in B cells[40,41], providing a potential molecular connection with integrin activity (Fig. 2a). Furthermore, both proteins were found to mediate BCR-controlled adhesion in other B cell lines as well.

Having identified several genes involved in (proximal) BCR signaling and BCR-controlled integrin-mediated adhesion, our results provide targets for therapeutic intervention of CLL, MCL, and WM, which rely on BCR-controlled adhesion of the malignant cells for their retention and survival in lymphoid tissues. More specifically, these targets might have clinical potential in

**Fig. 4 Regulators of proximal BCR-integrin signaling. a** Log2 fold changes of the guides (blue lines) targeting all significant proximal BCR-integrin signaling hits. Significant drop-out genes are identified from the αIgM/PMA statistics of the Namalwa screen. The RRA scores are the αRRA$_{depletion}$ scores. The graphs show the results for αIgM-induced adhesion relative to pre-adhesion (left), PMA-induced adhesion relative to pre-adhesion (middle), and αIgM-induced adhesion relative to PMA-induced adhesion (right). **b** The median fold changes of the 8 guides from genes represented in the kinome-centered Brunello library from the PMA-induced adhesion arm were plotted against the ones from the αIgM-induced adhesion arm of the Namalwa screen performed with 3 independent replicates. The significant genes are shown in black, and the size of the dots corresponds to the inverse of the αRRA$_{depletion}$ scores. **c** The fold changes of the individual guides from the Namalwa screen (DESeq2 calculates the maximum likelihood estimation (MLE) to determine the fold changes from 3 independent replicates) were plotted against the average adhesion level of the validation assays (*n* is shown in Fig S6B). The dark grey area represents the 95% confidence interval, and the light grey area the 95% prediction interval. sgNT represents a non-targeting guide, and sgNEG a guide against the non-essential/non-expressed gene BRDT. **d** Namalwa and JeKo-1 cells with indicated knockouts or 1 h pretreatment with indicated drugs, were stimulated with αIgM or PMA, and allowed to adhere to fibronectin-coated surfaces. The average inhibition of adhesion (Δ adhesion) compared to DMSO treatment or control guide (sgNEG) of independent adhesion assays is presented in a heatmap (*n* and *P* values are shown in Fig S6B). Source data are provided as a Source data file.

patients with primary (intrinsic) ibrutinib-resistance or, as combination therapy with ibrutinib, to further improve the mobilization of lymphoid tissue resident malignant B cells and/or to prevent or overcome the development of secondary (acquired) ibrutinib resistance.

Although the fold changes of individual guides are not as high as in most lethality screens[24], where each cell division causes an exponential increase of the fold-change between drop-out and non-functional guides, DESeq2 followed by αRRA was still very robust in identifying the significant hits. In addition, the median fold change of the guides of a particular gene was highly predictive for the degree of inhibition in validation assays. A confounding factor in the CRISPR adhesion screen is that viability genes will affect adhesion in an indirect manner. In general, apart from lethality screens, this will hamper interpretation in all functional CRISPR screens in which the function may be compromised by impaired viability of the cells, such as signaling screens. To overcome this problem, these genes can be filtered out by including an additional experimental arm with a distal stimulus - in our case the arm with PMA-stimulation - and calculating the fold changes with this sample as reference. If this is not possible, we recommend to perform a lethality screen in parallel to identify the genes critical for cell viability.

The screening technology described here can be easily adapted to study other cell types, cell adhesion-promoting stimuli (e.g., chemokines, growth factors, or cytokines), cell adhesion molecules (e.g., other integrins/subunits, cadherins, or selectins) and their ligands (either cellular or extracellular matrix), but also enables identification of cell adhesion-regulatory signaling pathways underlying unspecified cell–cell interactions in a co-culture setting. For example, we have recently employed this platform to conduct successful loss-of-adhesion CRISPR screens for integrin α$_4$β$_1$-mediated adhesion of multiple myeloma cells to VCAM-1 upon stimulation of the chemokine receptor CXCR4 with the chemokine CXCL12.

In conclusion, we have developed a powerful and flexible loss-of-adhesion CRISPR screening platform to identify and molecularly dissect the signaling pathway(s) underlying cell adhesion, including BCR-controlled integrin-mediated adhesion. This pathway is crucial for antigen-controlled retention of the malignant cells in their protective and growth-promoting microenvironment in various types of B cell malignancies, such as CLL, MCL, and WM[9–13]. Similar to BTK inhibitors like ibrutinib, therapeutic inhibitors against additional targets in the antigen/BCR-integrin axis may provide great clinical efficacy in lymphoma and leukemia patients.

## Methods

**Materials**. The following reagents were used in this study. Plasmids: psPAX2 (Addgene #12260), pMD2.G (Addgene #12259), pGFP-N3 (Clontech), lenti-Cas9-BSD (Addgene #52962), Cas9-reporter (Addgene #67980), lenti-Guide-Puro Brunello kinome library (Addgene #1000000082). Antibodies: mouse anti-Cas9 (Clone 7A9-3A3, #ab191468, Abcam), phycoerythrin-conjugated goat F(ab')$_2$ anti-mouse IgG1 (Polyclonal, #1072-09, Southern Biotech), goat F(ab')$_2$ anti-human IgM LE/AF (Polyclonal, #2022-14, Southern Biotech), mouse anti-CRKL (Clone 32H4, #3182, Cell Signaling Technologies), rabbit anti-PAK2 (Polyclonal, #2608, Cell Signaling Technologies), rabbit anti-PKM2 (Polyclonal, #3198, Cell Signaling Technologies), rabbit anti-PTK2B (Polyclonal, #3292, Cell Signaling Technologies), mouse anti-BTK (Clone 53/BTK, #611117, BD Transduction Laboratories), mouse anti-β-actin (Clone AC15, #A1978, Sigma-Aldrich), mouse anti-β-tubulin (Clone D66, #T0198, Sigma-Aldrich), goat anti-mouse-HRP (Polyclonal, #P0448, DAKO) and mouse anti-rabbit-HRP (Polyclonal, #P0260, DAKO). Pharmacological inhibitors: ibrutinib, idelalisib, FRAX597, and YM201636 (Selleck Chemicals), R406 and MK-2206 (MedChem Express), PD-98059 and wortmannin (Sigma-Aldrich), and AZ13705339 (Tocris Bioscience). Miscellaneous: BsmBI (New England Bio-Labs), puromycin (Sigma-Aldrich), blasticidin S hydrochloride (ThermoFisher), bovine serum albumin fraction V (Roche), human plasma fibronectin, and poly-L-lysine hydrobromide (Sigma-Aldrich), linear polyethylenimine 25 K (Polysciences), and ethylenediamine-tetraacetic acid (Merck).

**Cell culture**. Namalwa (DSMZ #ACC24), JeKo-1 (DSMZ #ACC553) and BCWM.1 (kindly provided by S.P. Treon) were cultured in IMDM supplemented with 10% fetal bovine serum (HyClone), glutamine, and penicillin/streptomycin. HEK-293T/17 (ATCC #CRL11268) was cultured in DMEM supplemented with 10% fetal bovine serum, glutamine and penicillin/streptomycin. Authentication of the cell lines and their derived Cas9 clones was performed by STR profiling (Powerplex 16, Promega) followed by DSMZ's online STR matching analysis. Cell cultures were regularly tested negative for mycoplasm contamination[42]. Cas9-expressing cells are readily available upon request.

**Library amplification**. In all, 1 μg plasmid library was electroporated (Gene Pulser, Biorad) in 1 vial ENDURA electrocompetent cells (Lucigen) and recovered in 1 ml SOC medium for 1 h at 37 °C. From 999 μl, 400 ml LB with ampicillin was inoculated, and 4 midipreps (Macherey-Nagel) were performed and pooled. From 1 μl, consecutive dilutions ($10^{-3}$–$10^{-6}$) were plated. Next day, the colonies were quantified. Our rule of thumb is to have at least 1000 times more colonies than the number of unique guides present in the library (which is for kinome-centered Brunello library[28] $> 6.2 \times 10^6$ colonies), to maintain the complexity of the library. This library includes 6204 guides which target 763 kinase(-related) genes, of which 518 protein kinases, 20 lipid kinases, and 225 other kinases or kinase-related genes (which contain ATP binding domains), each covered by 8 guides, as well as 100 non-targeting control guides.

**Cloning**. LentiGuide-Puro was digested with BsmBI and purified by gel extraction (Macherey-Nagel). Single stranded oligos (Sigma) containing the guide sequence and 25nt overlap with digested LentiGuide-Puro on each side (Supplementary Table 1) were cloned with the Gibson cloning kit (NEB). Chemo-competent Stbl3 cells (Thermofisher) were transformed with the Gibson products by heat shock. Upon minipreps (Macherey-Nagel), the plasmids were verified by Sanger sequencing (ABI).

**Virus production and transduction**. HEK-293T/17 cells were cultured in a 6-well plate to 80% confluency. In all, 1 μg psPAX2, 0.5 μg pMD2.G, 0.1 μg pGFP-N3, and 2 μg transfer plasmid were mixed with 8 μg polyethylenimine in 400 μl serum free DMEM, and after 15 min of incubation at room temperature this mixture was added to the cells. The next day the HEK cells were checked for GFP expression. If >50% of HEK cells were GFP positive, medium was refreshed, incubated for 16 h, filtered (0.45μm) and stored in −80 °C. For library transductions, HEK cells were grown in five T175 culture flasks, and the transfection mixture was scaled up relative to the surface area of the culture flasks.

In all, $0.5 \times 10^6$ cells in 2 ml IMDM were cultured for 24 h, and transduced with 100 µl (Namalwa) or 500 µl (JeKo-1 and BCWM.1) lentiviral supernatant. The next day medium was refreshed, and the transduced cells were selected with 10 µg/ml blasticin for 1 week, or with 2.5 µg/ml puromycin for 2–3 days. Only Namalwa-Cas9 cells were clonally sorted (Sony SH800) to obtain a clone with low background adhesion. All other transduced cells were used polyclonal to prevent clonal differences. The virus titer of the CRISPR library was first titrated. For screening, the virus titer for which the viability of the cells (determined by FSC/SSC on a Guava flow cytometer) was 30% relative to the viability of transduced cells without puromycin treatment (so multiplicity of infection (MOI) of 0.3), was used to have mainly cells with single virus integration[31] (which was typically 1:30 for transduction of Namalwa with lentiGuide-derived libraries).

**Cas9 intracellular FACS**. In all, $2 \times 10^5$ cells (Cas9-transduced and parental cells) were fixed (Fix buffer, BD Phosflow), permeabilized (Perm buffer III, BD Phosflow) and incubated with anti-Cas9 followed by PE-conjugated anti-mouse-IgG1. Stainings were measured on a FACScanto II flow cytometer system (BD Biosciences, San Jose, CA, USA) interfaced to FACS Diva software (v6.0), and analyzed with Flow Jo (v7.6.5).

**Cas9 reporter**. In all, $2 \times 10^5$ cells (Cas9-transduced and parental cells) were transduced with Cas9-reporter lentivirus (pKLV2-U6gRNA5(gGFP)-PGKBFP2AGFP-W)[29]. Cas9 activity results in loss of GFP expression, but not BFP expression. 6 days after transduction BFP and GFP expression were measured on a FACScanto II flow cytometer system (BD Biosciences, San Jose, CA, USA) interfaced to FACS Diva software (v6.0), and analyzed with Flow Jo (v7.6.5).

**Adhesion assay**. The cell-adhesion assays were performed essentially as described previously[10,27]. In detail, adhesion assays were done in triplicate on 96-well high binding plates (Greiner) coated overnight at 4 °C with PBS containing, 2.5 µg/ml fibronectin, or for 15 min at 37 °C with 1 mg/ml poly-L-lysine, and blocked for 1 h at 37 °C with 4% BSA in RPMI 1640. Cells were pretreated with 100 nM ibrutinib, 1 µM fostamatinib (R406). 100 nM wortmannin, 50 µM PD-98059, or 2.5 µM MK-2206 at 37 °C for 1 h in RPMI with 1% BSA. The inhibitors of the negative controls, PD-98059 and MK-2206, were used in concentrations which strongly block BCR-controlled ERK and AKT phosphorylation respectively[27,43]. Subsequently, cells were stimulated with either 100 ng/ml goat F(ab')₂ anti-human IgM, or 50 ng/ml phorbol-12-myristate-13-acetate (PMA), and $1.5 \times 10^5$ cells were immediately plated in 100 µl/well and incubated at 37 °C for 30 min. After extensive washing of the plate with RPMI containing 1% BSA to remove non-adhered cells, the adherent cells were fixed for 10 min with 10% glutaraldehyde in PBS and subsequently stained for 45 min with 0.5% crystal violet in 20% ethanol. After extensive washing with water, the dye was eluted in ethanol and absorbance was measured after 40 min at 570 nm on a spectrophotometer (ClarioStar). Absorbance due to nonspecific adhesion, as determined in wells coated with 4% BSA—which was always <10% of the absorbance of anti-IgM-stimulated cells—was subtracted. Maximal adhesion (100%) was determined by applying the cells to wells coated with poly-L-lysine, without washing the wells before fixation. Adhesion of the non-pretreated anti-IgM stimulated cells was normalized to 100% and the bars represent the mean + SEM of seven independent experiments, each assayed in triplicate. The data points are presented in a paired way from left to right. The plots were generated in Python (Anaconda3).

**Screening**. The screen was performed with 3 independent replicates, in which the pre-adhesion, anti-IgM and PMA adhesion samples were paired. To maintain the complexity of the library, the minimal amount of cells should be at least 1000 times the number of unique guides present in the library (which is for kinome-centered Brunello library[28] > $6.3 \times 10^6$ cells). Per replicate $60 \times 10^6$ Namalwa cells were transduced with the lentiGuide Brunello kinome-centered library at a multiplicity of infection (MOI) of 0.3. After puromycin selection (untransduced cells at the same cell density were included as puromycin control), the cells were cultured till > 90% of the cells were viable, and the number of viable cells was > $80 \times 10^6$ (6 days after puromycin selection for Namalwa). Cell pellets of $12 \times 10^6$ cells were stored in −20 °C as pre-adhesion samples. In all, $30 \times 10^6$ cells were stimulated with either 100 ng/ml goat F(ab')₂ anti-human IgM, or 50 ng/ml phorbol-12-myristate-13-acetate (PMA) in IMDM without serum at a cell density of $1.5 \times 10^6$/ml, and 100 µl/well ($1.5 \times 10^5$ cells) were immediately plated in fibronectin-coated wells and incubated at 37 °C for 30 min (two 96 well plates per sample), with an expected adhesion rate of 50%. After extensive washing of the plate with IMDM containing 1% BSA to remove non-adhered cells, the adherent cells were detached with 5 mM EDTA/PBS, and cell pellets were stored in −20 °C. Genomic DNA was isolated from the cell pellets (Quick DNA miniprep kit, Zymo Research), and DNA amounts were determined by Qubit fluorometry. Our recovery was typically 50–70%, with the rule of thumb of 6 pg of genomic DNA per cell.

**PCR and next-generation sequencing**. The PCRs were performed essentially as described[30,31]. In detail, from each sample 40 µg genomic DNA (~$6.5 \times 10^6$ cells) was used for 40 PCR reactions (in total 2 ml) with Phusion high fidelity polymerase (Thermo Fisher). The first PCR contained 12 barcodes in the forward primers,[30,31]

and Illumina sequencing primer sequences in both primers, and takes 21 cycles. The 40 PCR reactions were pooled, and from 2 µl of this pool a second PCR of 15 cycles was performed with primers containing the Illumina P5 and P7 sequences and a barcode in the reverse primer.[30,31] For sequencing of the plasmid library, 1 ng plasmid was used in single PCR reaction (50 µl) for the first PCR. PCR products were analyzed on an agarose gel and purified on a column (Gel and PCR purification kit, Macherey-Nagel). DNA concentrations of purified PCR products were determined by Qubit fluorometry. Equimolar amounts of DNA were pooled and subjected to Illumina next-generation sequencing. A HiSeq2500 lane contains usually ~$200 \times 10^6$ reads, and with the Brunello kinome-centered library (6204 unique guides) a maximum of 30 samples could be pooled.

**CRISPR screen analysis and statistics**. From the fastq file (SRR16971271) a count table was generated with Perl. In brief, from every read the barcode and guide sequence was determined. The reads were sorted on barcode (sample) and unique guides were counted. These count table was aligned to the reference table of the library. For barcode determination one mismatch was allowed, and for guide sequence determination no mismatches were allowed, but a shift of maximal 3 nucleotides was allowed (which are caused by indels in the synthetic PCR primers).

In R, the DESeq2 pipeline[32] (including shrinkage of non-informative fold changes and local fit to estimate the dispersion) was followed to obtain statistics on guide level from the count table. From these results the replicate plots, MA plots, and guide fold change plots were generated. To obtain statistics on gene level, αRRA (from the MAGeCK pipeline[33], with guide rankings on fold changes, and α-criterion on p-values) was performed on the DESeq2 data. In several screens, we obtained the most robust results with these settings. Based upon the median fold change of the guides per gene from the DESeq2 data, and from the αRRA depletion and enrichment scores (−log10 rho (depleted or enriched)), the volcano plots were generated. Plots were generated in Python (Anaconda3). (Strong) viability genes were determined by comparing read counts of the pre-adhesion samples with the distribution of the library (significant: FDR < 0.1 DESeq2 followed by αRRA). Genes with less than 1 transcript per million (TPM) were defined as non-expressed genes in Namalwa. RNAseq data from Namalwa (Broad Institute, SRR8615345), was downloaded from NCBI. The sra file was converted to a fastq file with fastq-dump from the SRA Toolkit (https://github.com/ncbi/sra-tools, reads were trimmed with seqtk (https://github.com/lh3/seqtk), mapped to the genome with HiSat2[44], mapped reads (sam file) were assembled and counted with Rsubread (featureCounts)[45]. Gene annotations were determined with biomaRt[46], and gene counts were normalized to transcripts per million (TPM).

**Western blot**. Protein lysates (whole-cell extracts in RIPA-buffer) were separated on Bolt™ 4-12% Bis-Tris Plus gels (Invitrogen) and subsequently blotted to a PVDF-membrane. The blots were blocked in 5% milk/TBST and incubated with indicated primary antibodies. Primary antibodies were detected with goat anti-mouse-HRP or mouse anti-rabbit-HRP, followed by detection using Pierce™ ECL Western Blotting Substrate (Thermo Scientific). All antibodies were used in an 1:1000 dilution, except for β-actin, which was used 1:5000.

**Statistics and reproducibility**. Adhesion assays were analyzed in R by two-way analysis of variance (ANOVA) followed by Tukey honestly significant difference (HSD) post hoc test, using a paired design (res_aov <- aov(adhesion ~ stimulus * inhibitor + replicate, data = data); res_thsd <- TukeyHSD(res_aov); padj <- res_thsd $'stimulus:inhibitor'), and the adjusted P values are shown. ANOVAs are one-tailed by its nature, and Tukey HSD test is two-tailed.

No statistical method was used to predetermine sample size. No data were excluded from the analyses. The experiments were not randomized. The investigators were not blinded to allocation during experiments and outcome assessment.

**Reporting summary**. Further information on research design is available in the Nature Research Reporting Summary linked to this article.

## Data availability

The raw CRISPR screen data are available in the NCBI SRA database under the accession number "SRR16971271". RNAseq data from Namalwa cells was taken from "SRR8615345" (Broad Institute). All other relevant data supporting the key findings of this study are available within the article and its Supplementary Information files or from the corresponding author upon reasonable request. Source data are provided with this paper.

## Code availability

The Perl, R, and Python scripts are available at the public GitHub repository (https://github.com/MFMdeRooij/CRISPRscreen / https://doi.org/10.5281/zenodo.6342853).

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

## Acknowledgements

The authors thank Cor Lieftink, Roel Kluin, and Jan Koster for bioinformatics hints & tips and Marjolein Lansbergen and Chayenne Veerman for generating some of the knockout cells. This work was supported by grants of Dutch Cancer Society (#7873, #10275), Lymph&Co, and the International Waldenstrom's Macroglobulinemia Foundation (IWMF)/Leukemia & Lymphoma Society (LLS) to M.S. and a grant of the Dutch Cancer Society (#12539) to R.L.B.

## Author contributions

M.F.M.d.R. designed the research, performed experiments, performed bioinformatics, analyzed the data, designed the figures, and wrote the manuscript; Y.J.T. designed the research, performed experiments, and analyzed the data; N.S. performed experiments; R.L.B. and S.T.P. cosupervised parts of the study; and M.S. supervised the study and wrote the manuscript. All authors reviewed the manuscript.

## Competing interests

The authors declare no competing interests.
