## [Peer Review File · Nature Communications]

Reviewers' comments:

Reviewer #1 (Remarks to the Author):

This study presents a kinase-oriented screen intended to identify regulators of BCR-stimulated adhesion of lymphoma cells to fibronectin, as potential targets of inhibitors that could displace the malignant B cells from survival signals in the tissue. This is a well intentioned screen in an area of BCR signalling that has been relatively poorly explored. The approach is very promising. However, while the authors demonstrate that the screen worked very well in their chosen cell line, the data are rather preliminary. The authors do not actually show the full screen data, nor do they validate any of the novel genes. At the very least, some basic mechanistic information on the role of the potentially novel candidates in integrin-mediated adhesion should be provided for publication.

Specific major points:

1. The choice of the cell line should be better explained because Namalwa is a Burkitt lymphoma line, while the clinically significant dependence on adhesion mostly relates to CLL. Validations should be extended to more relevant lymphoma types.
2. The manuscript need to include raw screen data as well as the analysed results in full as supplementary files. There is no way to judge the screen just based on the (only partly unannotated) dot plots. Transparency about the full dataset will greatly help sharing information in this emerging field. Were the three replicates averaged?
3. Since the authors show that there is a correlation between genes affecting survival and genes affecting adhesion, full analysis of this effect for the novel candidates should be provided. Also, given that many of the BCR-dependent adhesion regulators are also important for cell growth in relevant models (typically Syk, Btk - although apparently not in Namalwa), focusing the screen purely on the adhesion results may miss some important targets that regulate both adhesion and survival.

Minor:

4. Fig 1A - what are the two linear fit models to the data?
5. In the text, description of results jumps back and forth between Figure 2 and 3 and 3 and 4. Better organisation of the figures reflecting main points could help. For example, one figure focusing on screen quality, while the next on specific results.
6. Testing library efficiency is best done by calculating gRNA depletion for known survival genes, rather than showing that some genes had statistically significant effect on survival (especially as it is not revealed what genes these are).
7. Displaying the results using different symbols for statistical significance (like triangles versus disks) makes the screen plots difficult to read. May be a more prominent difference, like a contrasting color would be better.
8. The scheme in Fig4a has too many different colors for the proteins and it is difficult to tell which of them relate to screen results. One color for the proteins not present in the library would simplify the appearance.

Reviewer #2 (Remarks to the Author):

The manuscript "A loss-of-adhesion CRISPR-Cas9 screening platform to identify cell adhesion-regulatory proteins and signaling pathways" by deRoos et al, describes a screening platform for regulators of stimulated cell adhesion. The assay described is relatively straight-forward utilizing a commercially available kinome-library of guides, a lymphoma cell line Nawalma, and two conditions, BCR stimulation and PMA stimulation. While the screen proves its potential by successfully identifying the known regulators of BCR -dependent integrin-mediated adhesion, the impact and novelty of the study seem rather modest. Most importantly, the authors do not provide the full dataset and nor do they present evidence of identification of any novel regulators of adhesion. If the purpose of the manuscript is solely technical, i.e. to describe the CRISPR screen, the novelty remains limited and more thorough work in validating different adhesion conditions or cell types, for instance, would be required, in my opinion, to merit publication at Nature Communications.

Specific comments:

The justification of using fibronectin remains unclear. Other integrin ligands might be more relevant for the model system. Also, why cell adherence without prior stimulus was not included in the screen?

Fig 4. The graph type should be revised to allow estimation of the experimental variation.

in Supp Fig 3: in the α IgM/PMA panel (right), LYN guides seem to be enriched. This notion should be commented.

Full data, as well as a larger trimmed list of hits as part of the main figure, should be provided. The discussion states identification and verification of novel genes, as well as novel targets of clinical potential, without mentioning of what these genes are and how are they novel.

Minor comments:

The content of the library used seems mixed and should be better explained. If only kinases are targeted, why does the screen identify ACTR2?

Fig 3: Difficult to distinguish between the individual gene symbols in the charts. Also the embedded texts are difficult to read.

Usage of nomenclature should be checked, such as ACTR2 vs ARP2/3.

For the sake of open science, the scripts generated for the data analysis would be nice to be shared (such as for "From the fastq files a count table was generated with Perl"). Also the citations and/or links to the used programs should be provided (such as fastq-dump" and "seqtk).

Pieta Mattila, University of Turku, Finland

Reviewer #3 (Remarks to the Author):

In this technical report, de Roos et al. present a focused CRISPR screen to identify kinases involved in BCR-dependent cellular adhesion. The authors postulate that adhesion of malignant B cells in their microenvironment is the major consequence of oncogenic BCR signaling (as opposed to survival signaling) and that effective use of BCR targeted therapies requires loss of cellular adhesion. They further imply that therapy resistance is due to ineffective inhibition of adhesion and thus represents a critical question with clinical relevance. De Roos et al. engineered a Burkitt lymphoma (BL) model with constitutive Cas9 expression, select an exceptional cutting BL clone and perform a novel CRISPR

screen for cellular adhesion. They use anti-IgM or PMA stimulation to induce cell adhesion and compare the sgRNA frequencies in pre-stimulated control cells versus stimulated and still adhered test cells. The authors find BTK, SYK and PI3K all score well in their screen and show that inhibitors of these proteins promote a loss anti-IgM-mediated cell adhesion. Finally, they highlight how other kinases known to be involved in BCR signaling score in their assay.

This work expands upon the authors' previous studies examining the role of the BCR, and BTK/PI3K in particular, in mediating cellular adhesion in chronic lymphocytic leukemia (CLL), mantle cell lymphoma (MCL), and Waldenström's Macroglobulinemia (WM) in which the authors have shown that ibrutinib treatment inhibits adhesion in these diseases (Blood. 2012 Mar 15;119(11):2590-4; Blood. 2013 Oct 3; 122(14): 2412–2424; Blood. 2015 Apr 2; 125(14): 2306–2309.; Leukemia. 2016 Feb;30(2):337-45.; Haematologica. 2016 Mar; 101(3): e111–e115). The question of what mechanisms control retention of malignant cells in their niche is interesting and potentially therapeutically relevant. The use of an unbiased CRISPR screen for BCR-mediated adhesion is novel and will be of general interest to the lymphoma and immunology communities. In general, the experiments are well performed, and the data is fairly well presented.

Although the authors undertook an innovative approach to understanding this phenomenon, the effect sizes are modest and the authors' choice to perform the screen (and replicate the findings,) in a single BL cell line strongly decrease the enthusiasm for this manuscript. Of all the lymphoid malignancies, BL have only a small contribution of non-malignant stromal cells to the tumor mass. Diffuse large B cell lymphoma would most likely be a better disease in which to model stromal interactions.

The use of the very strong, non-physiologic, stimuli (anti-IgM and PMA) raises further questions concerning the relevance of the findings to the type and quality of BCR signaling present in lymphomas in which ibrutinib is either approved (MCL, WM, CLL) or under investigation for treatment in the ABC subtype of DLCL. By contrast, ibrutinib appears to have little efficacy in GCB DLBCL, which, like BL, is derived from germinal center B cells. While some BL cell lines use the BCR to engage the PI3 kinase pathway, they do not require BTK or NF- κ B activation. Other BLs do not require the BCR at all for survival, including the particular BL model cell line that the authors chose, Namawala. Given the authors' previous investigations into this adhesion in other B cell malignancies, it is surprising that they chose to highlight BL to demonstrate that BTK is required for anti-IgM-induced adhesion, which itself is finding that has been described by these investigators previously. They offer no additional mechanistic data as to how BTK regulates adhesion and decided not to pursue their top hit, PAK2, which has almost no known role in BCR signaling. Taken together, the authors have not presented a compelling argument to support relevance of their function screen to human lymphomas and their treatment.

Minor points

"Previously, we and others demonstrated that targeting of BCR-controlled integrin-mediated adhesion underlies the clinical efficacy of BCR signalosome inhibitors.

-This statement is at odds with other published data showing cytotoxicity with these inhibitors.

"The currently most common clinically applied BCR-signalosome inhibitors are the BTK inhibitors ibrutinib and acalabrutinib. However, upon prolonged treatment, patients develop resistance against these inhibitors, mainly due to mutation in BTK or its substrate PLC γ 214-19."

-BTK and PLC γ 2 mutations are associated with BTK inhibitor resistance, but mutations are sporadic and highly dependent on the subtype of B cell malignancy. Perhaps the authors should tone down this statement.

Figure 1 contains no data

Figure 2 largely contains supportive data showing that the CRISPR screen technically worked. These data belong in the supplement.

Figure 4A is a nice depiction of BCR signaling, but does not convey any information from this line of investigation. Could the authors redesign this to overlay the results of the screens? Consider moving this to Figure 5 for better comprehension of the screen results.

Figure 4B. This data, or a very similar experiment, has been published previously by this group (Blood 2012).

The methods mention an additional cell line – Jeko – was used in this study, but no data is presented. Are the text and figure legends incorrect, or the methods?

Typos:

2log scale

death cells

as [^]a positive regulator

Rebuttal: NCOMMS-20-12140-T, by de Rooij et al., entitled "A loss-of-adhesion CRISPR-Cas9 screening platform to identify cell adhesion-regulatory proteins and signaling pathways"

Reviewers' comments:

Reviewer #1 (Remarks to the Author):

This study presents a kinase-oriented screen intended to identify regulators of BCR-stimulated adhesion of lymphoma cells to fibronectin, as potential targets of inhibitors that could displace the malignant B cells from survival signals in the tissue. This is a well intentioned screen in an area of BCR signalling that has been relatively poorly explored. The approach is very promising. However, while the authors demonstrate that the screen worked very well in their chosen cell line, the data are rather preliminary. The authors do not actually show the full screen data, nor do they validate any of the novel genes. At the very least, some basic mechanistic information on the role of the potentially novel candidates in integrin-mediated adhesion should be provided for publication.

We thank the reviewer for reviewing the manuscript.

Please allow us to point out that the original manuscript was a method paper, submitted as a Technical Report.

We have now included all requested information, as detailed below.

Specific major points:

1. The choice of the cell line should be better explained because Namalwa is a Burkitt lymphoma line, while the clinically significant dependence on adhesion mostly relates to CLL. Validations should be extended to more relevant lymphoma types.

We have now explained the choice of Namalwa as a model cell line (page 5 line 81-84).

Furthermore, we have now also included new data on the role of the previously studied genes (BTK, PI3K δ and SYK) as well as novel genes (PAK2, PTK2B, PKM and CRKL) in the BCR-controlled integrin-mediated adhesion of the mantle cell lymphoma cell line JeKo-1 (and the Waldenstrom macroglobulinaemia cell line BCWM.1) (Fig. 3D and Suppl. Fig. 6).

2. The manuscript need to include raw screen data as well as the analysed results in full as supplementary files. There is no way to judge the screen just based on the (only partly unannotated) dot plots. Transparency about the full dataset will greatly help sharing information in this emerging field. Were the three replicates averaged?

We have now included all raw screen data in the source data file, and all analyzed results in full (e.g., see Fig. 3, Supplemental Fig. 5). Furthermore, all raw data will be publically accessible at NCBI's sequence read archive (SRR16971271) after publication. Yes, the 3 replicates were averaged: they were used in the DESeq2 pipeline, which requires at least 2 replicates for its statistics.

3. Since the authors show that there is a correlation between genes affecting survival and genes affecting adhesion, full analysis of this effect for the novel candidates should be provided. Also, given that many of the BCR-dependent adhesion regulators are also important for cell growth in relevant models (typically Syk, Btk - although apparently not in Namalwa), focusing the screen purely on the adhesion results may miss some important targets that regulate both adhesion and survival.

We have now provided all data concerning lethality of the genes in Namalwa (Suppl. Figs. 2C,D and 4; and source data file).

Minor:

4. Fig 1A - what are the two linear fit models to the data?

We apologize for the omission of this information. The black line in current Suppl. Fig. 2A is the diagonal $y=x$, the other line is the linear regression curve. This information is now included in the figure legend.

5. In the text, description of results jumps back and forth between Figure 2 and 3 and 3 and 4. Better organisation of the figures reflecting main points could help. For example, one figure focusing on screen quality, while the next on specific results.

We apologize for this inconvenience and have reorganized several figures and panels to improve the readability.

6. Testing library efficiency is best done by calculating gRNA depletion for known survival genes, rather than showing that some genes had statistically significant effect on survival (especially as it is not revealed what genes these are).

We have now provided all information on the identity of the genes, including labeling the spots representing these genes in the figures (e.g., Figs. 1B, 3A-C, and Suppl. Fig. 2B-C).

7. Displaying the results using different symbols for statistical significance (like triangles versus disks) makes the screen plots difficult to read. May be a more prominent difference, like a contrasting color would be better.

We have now adjusted the symbols and colors to improve interpretation/visibility.

8. The scheme in Fig4a has too many different colors for the proteins and it is difficult to tell which of them relate to screen results. One color for the proteins not present in the library would simplify the appearance.

We have now adjusted the colors. For explanation of the colors, please see the Figure legend.

Reviewer #2 (Remarks to the Author):

The manuscript “A loss-of-adhesion CRISPR-Cas9 screening platform to identify cell adhesion-regulatory proteins and signaling pathways” by deRoos et al, describes a screening platform for regulators of stimulated cell adhesion. The assay described is relatively straight-forward utilizing a commercially available kinome-library of guides, a lymphoma cell line Nawalma, and two conditions, BCR stimulation and PMA stimulation. While the screen proves its potential by successfully identifying the known regulators of BCR -dependent integrin-mediated adhesion, the impact and novelty of the study seem rather modest. Most importantly, the authors do not provide the full dataset and nor do they present evidence of identification of any novel regulators of adhesion. If the purpose of the manuscript is solely technical, i.e. to describe the CRISPR screen, the novelty remains limited and more thorough work in validating different adhesion conditions or cell types, for instance, would be required, in my opinion, to merit publication at Nature Communications.

We thank the reviewer for reviewing of our manuscript.

We have now provided the full dataset, presented the evidence of identification of novel regulators of adhesion, and included analysis in another cell line, as detailed below.

Specific comments:

The justification of using fibronectin remains unclear. Other integrin ligands might be more relevant for the model system. Also, why cell adherence without prior stimulus was not included in the screen?

We have now justified the use of fibronectin on page 5-6, line 102-105.

Cell adherence without prior stimulus cannot be included because the unstimulated cells hardly adhere and are washed out. Therefore, for comparison, the pre-adhesion cells were used (still containing all gRNAs).

Fig 4. The graph type should be revised to allow estimation of the experimental variation.

The experimental variation is reflected by the error bars.

in Supp Fig 3: in the α IgM/PMA panel (right), LYN guides seem to be enriched. This notion should be commented.

This enrichment was not statistically significant.

Full data, as well as a larger trimmed list of hits as part of the main figure, should be provided. The discussion states identification and verification of novel genes, as well as novel targets of clinical potential, without mentioning of what these genes are and how are they novel.

We have now provided the full data (source data file), as well as a larger trimmed list (Figure 3A-C)

Furthermore, we have now presented the identity, verification and discussion of the novel genes (Fig. 3; Suppl. Fig. 4-6).

Minor comments:

The content of the library used seems mixed and should be better explained. If only kinases are targeted, why does the screen identify ACTR2?

The library is kinome-centered and does not exclusively include kinases. ACTR2 is included as it contains an ATP binding domain. This information is now provided at page 17 line 344-347.

Fig 3: Difficult to distinguish between the individual gene symbols in the charts. Also the embedded texts are difficult to read.

We apologize for this inconvenience and have adjusted the symbols, colors and fonts to improve interpretation/visibility/readability.

Usage of nomenclature should be checked, such as ACTR2 vs ARP2/3.

Regarding nomenclature we use either the gene name (*ACTR2*) or the name of the encoded protein (*ARP2/3*), as appropriate.

For the sake of open science, the scripts generated for the data analysis would be nice to be shared (such as for "From the fastq files a count table was generated with Perl"). Also the citations and/or links to the used programs should be provided (such as fastq-dump" and "seqtk).

In the original manuscript we provided a link to the scripts (including Perl and R) on github (<https://github.com/MFMdeRooij/CRISPRscreen>) and we have now included the github links <https://github.com/ncbi/sra-tools> and <https://github.com/lh3/seqtk> to the linux apps SRA toolkit/fastq-dump and seqtk.

Reviewer #3 (Remarks to the Author):

In this technical report, de Rooij et al. present a focused CRISPR screen to identify kinases involved in BCR-dependent cellular adhesion. The authors postulate that adhesion of malignant B cells in their microenvironment is the major consequence of oncogenic BCR signaling (as opposed to survival signaling) and that effective use of BCR targeted therapies requires loss of cellular adhesion. They further imply that therapy resistance is due to ineffective inhibition of adhesion and thus represents a critical question with clinical relevance. De Rooij et al. engineered a Burkitt lymphoma (BL) model with constitutive Cas9 expression, select an exceptional cutting BL clone and perform a novel CRISPR screen for cellular adhesion. They use anti-IgM or PMA stimulation to induce cell adhesion and compare the sgRNA frequencies in pre-stimulated control cells versus stimulated and still adhered test cells. The authors find BTK, SYK and PI3K all score well in their screen and show that inhibitors of these proteins promote a loss anti-IgM-mediated cell adhesion. Finally, they highlight how other kinases known to be involved in BCR signaling score in their assay.

This work expands upon the authors' previous studies examining the role of the BCR, and BTK/PI3K in particular, in mediating cellular adhesion in chronic lymphocytic leukemia (CLL), mantle cell lymphoma (MCL), and Waldenström's Macroglobulinemia (WM) in which the authors have shown that ibrutinib treatment inhibits adhesion in these diseases (Blood. 2012 Mar 15;119(11):2590-4; Blood. 2013 Oct 3; 122(14): 2412–2424; Blood. 2015 Apr 2; 125(14): 2306–2309.; Leukemia. 2016 Feb;30(2):337-45.; Haematologica. 2016 Mar; 101(3): e111–e115). The question of what mechanisms control retention of malignant cells in their niche is interesting and potentially therapeutically relevant. The use of an unbiased CRISPR screen for BCR-mediated adhesion is novel and will be of general interest to the lymphoma and immunology communities. In general, the experiments are well performed, and the data is fairly well presented.

Although the authors undertook an innovative approach to understanding this phenomenon, the effect sizes are modest and the authors' choice to perform the screen (and replicate the findings,) in a single BL cell line strongly decrease the enthusiasm for this manuscript. Of all the lymphoid malignancies, BL have only a small contribution of non-malignant stromal cells to the tumor mass. Diffuse large B cell lymphoma would most likely be a better disease in which to model stromal interactions.

We thank the reviewer for appreciating the relevance, novelty and quality of our study.

Regarding stromal cell contribution, as compared with BL (but also DLBCL), CLL and MCL would have been a better choice. In this respect, please note that we have now extended our findings to a MCL cell line Jeko-1 (Fig. 3D and Suppl. Fig. 6). As to the use of the BL cell line Namalwa, please see our response below.

The use of the very strong, non-physiologic, stimuli (anti-IgM and PMA) raises further questions concerning the relevance of the findings to the type and quality of BCR signaling present in lymphomas in which ibrutinib is either approved (MCL, WM, CLL) or under investigation for treatment in the ABC subtype of DLBCL. By contrast, ibrutinib appears to have little efficacy in GCB DLBCL, which, like BL, is derived from germinal center B cells. While some BL cell lines use the BCR to engage the PI3 kinase pathway, they do not require BTK or NF- κ B activation. Other BLs do not require the BCR at all for survival, including the particular BL model cell line that the authors chose, Namalwa. Given the authors' previous investigations into this adhesion in other B cell malignancies, it is surprising that they chose to highlight BL to demonstrate that BTK is required for anti-IgM-induced adhesion, which itself is finding that has been described by these investigators previously.

First allow us to emphasize that the use of anti-IgM is well-established as a tool to activate the BCR and BCR signaling, including for MCL and CLL cell lines and primary cells.

Our motivation to use the Namalwa cell line was not because it is a BL cell line, but because (1) it is an excellent model to study BCR-controlled integrin-mediated adhesion of (malignant) B cells in general (not specifically for BL), including the role of BTK and ibrutinib (Spaargaren et al., 2003 JEM; de Rooij et al., 2012), (2) it is highly susceptible to viral transduction, and (3) knockout of (proximal) BCR pathway

components does not compromise viability of Namalwa cells, unlike for example in ABC-DLBCL cell lines. This is a critical requirement for a successful loss-of-adhesion CRISPR screen, since dead cells cannot be stimulated to adhere; for this reason we also included an experimental arm of PMA-stimulated adhesion in our experimental set-up and advocate analysis of lethality in parallel to the adhesion screen to identify false positives (*i.e.*, lethal genes).

We have now further explained/emphasized these aspects in the text of the revised version (Page 5, line 81-84, and page 14, line 284-288).

They offer no additional mechanistic data as to how BTK regulates adhesion and decided not to pursue their top hit, PAK2, which has almost no known role in BCR signaling. Taken together, the authors have not presented a compelling argument to support relevance of their function screen to human lymphomas and their treatment.

We have now pursued other top hits, including PAK2, both in Namalwa and in an MCL cell line (Fig. 3 and Suppl. Fig. 6), and discussed these novel genes and insights in the context of BCR-controlled regulation of adhesion as well as therapeutic potential (page 9-11, and 12-13).

At this point, please allow us to emphasize that among the top 10 hits of our screen we identified two targets for drugs with FDA-approval for CLL and MCL, *i.e.*, BTK and PI3K. This, combined with the clarifications and the inclusion of a substantial amount of new data in this revised manuscript, will hopefully convince the reviewer of the relevance of our screen/data for human lymphoma (beyond BL!) and their treatment.

Minor points

“Previously, we and others demonstrated that targeting of BCR-controlled integrin-mediated adhesion underlies the clinical efficacy of BCR signalosome inhibitors.”

-This statement is at odds with other published data showing cytotoxicity with these inhibitors.

It is generally appreciated that the unprecedented clinical efficacy of ibrutinib in CLL and MCL is due to impaired cell adhesion (lymphoid organ retention) and not direct cytotoxicity (e.g., Burger and Wiestner, 2018, Nat Rev Cancer 18(3):148-167). This is nicely illustrated by the lymphocytosis observed upon treatment of patients, which is the result of mobilization of malignant cells from the lymphoid organs into the blood, as opposed to direct cell death! Notably, also *in vitro*, at clinically achievable (=relevant) concentrations (up to approx. 300 nM) ibrutinib is not cytotoxic to CLL and MCL cells (although in some cell lines proliferation is mildly affected). This in contrast to its cytotoxic effect in the low nM range in a subset of diffuse large B cell lymphoma (DLBCL), *i.e.*, in ABC subtype DLBCL with chronic BCR signaling. Unfortunately, in many preclinical studies ibrutinib has been applied in the μM range, causing off-target effects.

“The currently most common clinically applied BCR-signalosome inhibitors are the BTK inhibitors ibrutinib and acalabrutinib. However, upon prolonged treatment, patients develop resistance against these inhibitors, mainly due to mutation in BTK or its substrate PLC γ 214-19.”

-BTK and PLC γ 2 mutations are associated with BTK inhibitor resistance, but mutations are sporadic and highly dependent on the subtype of B cell malignancy. Perhaps the authors should tone down this statement.

We have now rephrased this statement (page 3, line 54-61). Please note, however, that BTK and PLCG2 mutations were detected in 57% and 13% of ibrutinib treated patients with progressive CLL, respectively (Quinquenel et al., 2019, Blood 134(7):641).

Figure 1 contains no data

We have now combined the procedure/concept of the CRISPR loss-of-adhesion screen (Fig. 1a) with the actual data obtained by the CRISPR screen, presented both at guide (MA plots) and gene level (volcano plots) (Fig. 1b).

Figure 2 largely contains supportive data showing that the CRISPR screen technically worked. These data belong in the supplement.

We prefer to present these data in the core of the manuscript, as it demonstrates the feasibility and validation of the screen. Please note that the manuscript was considered to be a Technical report.

Figure 4A is a nice depiction of BCR signaling, but does not convey any information from this line of investigation. Could the authors redesign this to overlay the results of the screens? Consider moving this to Figure 5 for better comprehension of the screen results.

We thank the reviewer for the advice and have now redesigned this figure to better highlight genes identified in the screen.

Figure 4B. This data, or a very similar experiment, has been published previously by this group (Blood 2012).

This is indeed a similar experiment, now also including an AKT inhibitor. These results are presented as positive controls for the CRISPR screen: to illustrate and (experimentally) support the validity of our CRISPR screen we chose to conduct these new experiments with all relevant inhibitors for this manuscript, rather than merely referring to previously reported data for most inhibitors used.

The methods mention an additional cell line – Jeko – was used in this study, but no data is presented. Are the text and figure legends incorrect, or the methods?

JeKo-1 was used in the experiments presented in the Supplemental Figure 2 of the original manuscript.

Typos:

2log scale

death cells

as ^a positive regulator

We thank the reviewer for pointing out these typo's and have corrected them.

REVIEWERS' COMMENTS

Reviewer #1 (Remarks to the Author):

This is a much-improved manuscript. In general, the authors responded to my comments and addressed them satisfactorily. I only have minor comments on the presentation of the data:

1. In the abstract and intro, CLL is talked about as lymphoma. Is this a standard in the field? I suggest referring to it as leukaemia.
2. Fig 1b and Supplementary 1 d-c. The legend does not fully match the figure. There are open triangles in the legend which do not appear in the plots but are instead closed triangles. Also, triangles as symbols are too big and the data are too squished (even overlapping labels) to be appreciated.
3. In the signalling scheme of Fig 2a or in its legend, it might be good to indicate which genes are underlying the bubbles for PI3K and PKC (there are multiple genes in these families).
4. Fig 2B the legends indicate a black circle for the gene of interest, but I think the data are actually in black triangles. There are also blue and red triangles, unclear what they are.
5. P110delta is dismissed in the text as being redundant, but actually a PIK3CD is a hit of the anti-IgM screen with $FDR < 0.1$
6. Supplementary Fig 6 b, c statistics are missing in some panels. If the data are not significant, this should be indicated.

Reviewer #2 (Remarks to the Author):

In their manuscript (submitted as a technical report), "A loss-of-adhesion CRISPR-Cas9 screening platform to identify cell adhesion-regulatory proteins and signaling pathways" Rooij et al. setup a functional CRISPR screen for loss-of-adhesion. Malignant B cells often require a specific microenvironment and if the adhesion properties are perturbed, the cells are forced to leave their microenvironment inducing cell death. New therapies are designed to target these cellular pathways and this is also the motivation of this study. At the same time the study provides as a nice example of a CRISPR screen to study cell adhesion and the platform could be relatively easily adaptable also for other cellular systems. The authors have substantially improved the manuscript and now providing a nice demonstration of the power of the screen by improved analysis, validation of the selected hits also using another cell line. I do not have major concerns.

The full dataset is appended to the manuscript but it is not mentioned in the main text nor there is any explanation/legend to guide the reader.

The two first panels in 3a appear to be named wrongly.

It is highly questionable to call Arp2/3 as "kinase-related". This phrase (line 241) should be revised for example by focusing to the ATPase activity if that is the underlying reason this protein is included in the library.

Reviewer #4 (Remarks to the Author):

I have been asked to assess only whether the authors have sufficiently addressed the points raised by

reviewer 3 in the primary submission process. Assuming that I have had access to all of this reviewer's comments (which I believe to be the case) my impression is that the authors have done sufficient additional work and modifications to the manuscript to justify publication. As they point out, this was a Technical report and while that does not imply a lowering of any standards I would suggest that it would mean that, for example, a full molecular mechanistic description of a process would not be an appropriate demand to make of such a publication. To this end I believe the authors have now included all screen data, provided data on validation of additional hits from the primary screen and conducted some validation in alternative B cell systems all of which increase the confidence in the technical nature of the screen. They have acknowledged the limitations of the interpretation of some of the data but to some extent, it is for the rest of the scientific community to fully address the generality of the observations recorded here. The revised manuscript I read would, I feel, be of a sufficient quality to merit publication.

REVIEWERS' COMMENTS

Reviewer #1 (Remarks to the Author):

This is a much-improved manuscript. In general, the authors responded to my comments and addressed them satisfactorily. I only have minor comments on the presentation of the data:

We thank the reviewer for the positive response.

1. In the abstract and intro, CLL is talked about as lymphoma. Is this a standard in the field? I suggest referring to it as leukaemia.

Indeed, biologically CLL is/should be regarded as a lymphoma, because the malignant cells exclusively survive and proliferate in the lymphoid tissues, whereas cells in the circulation are not dividing; however, since clinically this terminology is not generally accepted, we have now changed “lymphoma” to “B-cell malignancies”, which covers both lymphoma and leukemia. (Page 2 line 23 and 31, page 3 line 38-39, page 4 line 59, page 13 line 276, page 14 line 306, and page 15 line 309).

2. Fig 1b and Supplementary 1 d-c. The legend does not fully match the figure. There are open triangles in the legend which do not appear in the plots but are instead closed triangles. Also, triangles as symbols are too big and the data are too squished (even overlapping labels) to be appreciated.

The colors were used to indicate the type of gene (guide), whereas the shapes reflected the statistical significance. We have now improved and clarified the legend.

3. In the signalling scheme of Fig 2a or in its legend, it might be good to indicate which genes are underlying the bubbles for PI3K and PKC (there are multiple genes in these families).

We have now indicated the individual genes.

4. Fig 2B the legends indicate a black circle for the gene of interest, but I think the data are actually in black triangles. There are also blue and red triangles, unclear what they are.

As indicated at point 2, the colors indicated the type of gene (guide), and the shape referred to the significance. We have now improved the legend.

5. P110delta is dismissed in the text as being redundant, but actually a PIK3CD is a hit of the anti-IgM screen with $FDR < 0.1$

We thank the reviewer for pointing this out, enabling us to clarify this issue.

In order to compensate for any toxicity, we focused our analysis to the IgM/PMA comparison.

PIK3CD had an α IgM/input with $fdr=0.094$, an PMA/input $fdr=0.104$, but α IgM/PMA $fdr=0.381$, and the guide distribution follows the guide distribution of viability genes. So indeed, with a cut-off at $FDR < 0.1$, it is a hit for anti-IgM over input, but not for α IgM/PMA. To clarify this we have now added “in the α IgM/PMA comparison” to the statement of PIK3CD not being significant. (page 8, line 152).

6. Supplementary Fig 6 b, c statistics are missing in some panels. If the data are not significant, this should be indicated.

In compliance with the reviewer and the author instructions of Nature Communications, we have now presented the p-values in all graphs.

Reviewer #2 (Remarks to the Author):

In their manuscript (submitted as a technical report), “A loss-of-adhesion CRISPR-Cas9 screening platform to identify cell adhesion-regulatory proteins and signaling pathways” Rooij et al. setup a functional CRISPR screen for loss-of-adhesion. Malignant B cells often require a specific microenvironment and if the adhesion properties are perturbed, the cells are forced to leave their microenvironment inducing cell death. New therapies are designed to target these cellular pathways and this is also the motivation of this study. At the same time the study provides as a nice example of a CRISPR screen to study cell adhesion and the platform could be relatively easily adaptable also for other cellular systems. The authors have substantially improved the manuscript and now providing a nice demonstration of the power of the screen by improved analysis, validation of the selected hits also using another cell line. I do not have major concerns.

We thank the reviewer for the positive response.

The full dataset is appended to the manuscript but it is not mentioned in the main text nor there is any explanation/legend to guide the reader.

We have now added the info sheet in the source data file.

The two first panels in 3a appear to be named wrongly.

We thank the reviewer for pointing this out. Upon rearrangement of the graphs, the final version was not saved well. This mistake is now corrected.

It is highly questionable to call Arp2/3 as “kinase-related”. This phrase (line 241) should be revised for example by focusing to the ATPase activity if that is the underlying reason this protein is included in the library.

Indeed, we assume that is the case. Please note that this library was created by the Broad Institute (Doench et al Nature Biotechnology 2016).

We have now added “Note that ARP2/3 is not a kinase complex, but ARP2 contains an ATP-binding domain for its ATPase activity.” (page 12, line 240-241).

Reviewer #4 (Remarks to the Author):

I have been asked to assess only whether the authors have sufficiently addressed the points raised by reviewer 3 in the primary submission process. Assuming that I have had access to all of this reviewer's comments (which I believe to be the case) my impression is that the authors have done sufficient additional work and modifications to the manuscript to justify publication. As they point out, this was a Technical report and while that does not imply a lowering of any standards I would suggest that it would mean that, for example, a full molecular mechanistic description of a process would not be an appropriate demand to make of such a publication. To this end I believe the authors have now included all screen data, provided data on validation of additional hits from the primary screen and conducted some validation in alternative B cell systems all of which increase the confidence in the technical nature of the screen. They have acknowledged the limitations of the interpretation of some of the data but to some extent, it is for the rest of the scientific community to fully address the generality of the observations recorded here. The revised manuscript I read would, I feel, be of a sufficient quality to merit publication

We thank the reviewer for the positive response.